# FROM ATTACKS TO GUIDANCE:
# DIRECT OUTPUT CONTROL FOR CLASSIFIERS

## ABSTRACT

In high-risk domains such as autonomous driving and medical diagnosis, classifier misclassifications pose severe risks. Existing repair approaches fall into three categories: test-time adaptation (TTA), adversarial perturbation methods such as PGD and DeepFool, and counterfactual generation (CF). TTA and perturbation methods lack stability guarantees or irreparability diagnosis, while CF targets distributional plausibility rather than direct control. We propose *Direct Output Control (DOC)*, which repairs misclassifications by directly regulating the output distribution without changing model parameters. DOC defines the Fisher - Rao distance as a Lyapunov function, pulls back its gradient through the Jacobian pseudoinverse, and derives minimum-norm perturbations that monotonically reduce error. The framework generalizes to other metrics (e.g., $L_2$) and provides both a theoretical irreparability bound based on Jacobian singular values and inter-class margins, and an empirical diagnostic using Lyapunov decrease. On ImageNet-1k with ResNets and Vision Transformers, DOC outperforms TTA and perturbation methods in repair success while inducing smaller distortions, though at higher inference cost. Our contributions are: (1) a Lyapunov-control formulation with monotonic stability, (2) theoretical analysis including irreparability, minimum-norm, and natural gradient connection, (3) an empirical diagnostic via Lyapunov decrease, and (4) large-scale validation showing Pareto superiority in success - distortion trade-offs.

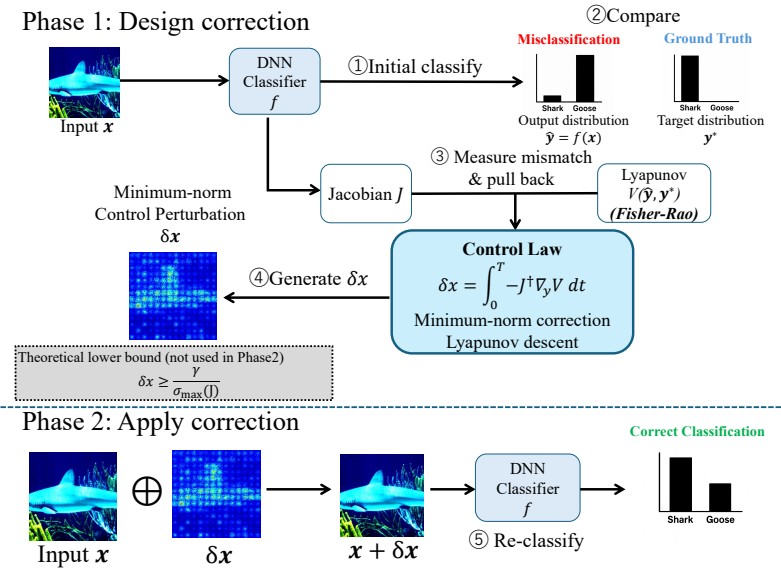

Figure 1: **Overview of Direct Output Control (DOC).** When a classifier produces a misclassification, a control law based on a Lyapunov function is used to derive a minimum-norm input perturbation $\delta x$, guiding the corrected input $x + \delta x$ toward the correct classification.

# 1 INTRODUCTION

In high-risk domains such as autonomous driving, medical image diagnosis, and industrial inspection, classifier misclassifications can directly lead to accidents, medical errors, and financial losses. In such cases, the ability to repair misclassifications at inference time with minimal intervention can immediately enhance safety and reliability. Minimal intervention is essential for the following reasons. First, smaller perturbations better preserve the original meaning and structure of the input, ensuring the interpretability and reliability of the repair. Second, limiting modifications to a local scope contains side effects, guaranteeing operational safety in the field. Third, the perturbation amount can be auditably recorded, facilitating compliance with regulations and accountability tracking.

Existing related methods can be broadly categorized into three groups. (i) Test-Time Adaptation (TTA): These methods update weights during inference to handle out-of-distribution shifts but impose a significant burden of modifying and re-validating the production model. (ii) Perturbation Methods Originating from Attacks (PGD, DeepFool): These methods can alter classifier outputs by adding perturbations to the input but are originally designed for attacks and lack safety-side guarantees. (iii) Counterfactual Generation (CF): This trend involves synthesizing data that yields correct classifications by modifying inputs, but its main focus is on verifying generation plausibility and distribution constraints, making it incomparable to frameworks like adversarial perturbation methods that "directly control the output." This study does not address CF and instead focuses on safe control design.

In this study, to overcome these challenges, we propose Direct Output Control (DOC), which controls the output distribution itself without modifying the trained model. DOC defines the distance between the model's output and a target output as a Lyapunov function and designs the input displacement in a way that guarantees its decrease. This ensures that the error is guaranteed to decrease at each step of inference, achieving stable repair. Furthermore, we introduce a mechanism to theoretically determine whether a repair is possible by calculating a lower bound based on Jacobian singular values and inter-class margins. This allows us to classify misclassifications into "reparable" and "fundamentally irreparable" cases, eliminating the black-box trial-and-error approach of conventional methods.

Moreover, DOC is structured to naturally derive the minimum-norm perturbation in the input space. This corresponds to the natural gradient in the output space, possessing optimality from both a control-theoretic and information-geometric perspective. As a result, the input modification required to repair a classification is minimized, significantly reducing visual distortions in image recognition tasks.

As a large-scale experiment, we compared DOC with adversarial perturbation methods on ResNet-50, ResNet-101(He et al. (2016)), ViT-B, and ViT-L(Dosovitskiy et al. (2021)) on ImageNet-1k(Deng et al. (2009)). A comparison with TTA is provided in the appendix. The results show that DOC consistently outperforms perturbation methods like PGD and DeepFool, as well as TTA, achieving a higher repair success rate while significantly suppressing input perturbations and perceptual distortions. Furthermore, we confirmed that the decreasing behavior of the Lyapunov function and diagnostics based on Jacobian properties can accurately identify irreparable samples.

The contributions of this research are summarized in the following four points:

1. Formulation of a new framework, "Direct Output Control," which treats the output distribution as the control target and guarantees monotonic error decrease via a Lyapunov function.
2. Theoretical analysis of its properties, including (i) a reparability condition based on Jacobian singular values and inter-class margins, (ii) the natural derivation of a minimum-norm perturbation, and (iii) its correspondence with the natural gradient in the output space.
3. An empirical diagnostic criterion using Lyapunov decrease, which enables practical detection of irreparability.
4. Large-scale validation on ImageNet-1k , demonstrating Pareto superiority in terms of both repair success rate and distortion.

Thus, DOC re-positions perturbation from an "attack" to a form of "guidance" for inference-time control, establishing a new paradigm that integrally enables stable repair and irreparability diagnosis.

The structure of this paper is as follows. §2 organizes the problem setup and notation. §3 formulates the framework and algorithm for Direct Output Control (DOC). Following §3, §4 provides a theoretical analysis of Lyapunov monotonic decrease, minimum-norm property, the relationship with the natural gradient, and irreparability diagnosis. §5 reviews related work and clarifies the positioning of our study. Subsequently, §6 validates the proposed method's repair performance, distortion, and diagnostic accuracy through large-scale experiments on ImageNet-1k, and §7 presents the experimental results. Finally, §8, §9, and §10 present the discussion, limitations and societal impact, and conclusion.

## 2 PROBLEM SETUP AND NOTATION

Let the input space be $\mathcal{X} \subseteq \mathbb{R}^d$ and the output space be the simplex over $K$ classes $\Delta^{K-1} = \{y \in \mathbb{R}_{\geq 0}^K \mid \mathbf{1}^\top y = 1\}$. For a pre-trained classifier $f : \mathcal{X} \to \Delta^{K-1}$, we denote the logits as $z(x) \in \mathbb{R}^K$ and the output as $y(x) = \mathrm{softmax}(z(x))$. Given an input $x$ and a target label $y^\star$ (usually a one-hot vector of the correct class), the objective is to bring $y(x)$ closer to $y^\star$ by applying a small perturbation $\delta x$.

**Error Metric**    The difference between outputs is measured by a distance function $D : \Delta^{K-1} \times \Delta^{K-1} \to \mathbb{R}_{\geq 0}$. We standardly use the Fisher–Rao distance
$$d_{\mathrm{FR}}(y, y^\star) = 2 \arccos\big(\langle \sqrt{y}, \sqrt{y^\star} \rangle\big)$$
and define the Lyapunov function as $V(x) = d_{\mathrm{FR}}(y(x), y^\star)^2$. (Results for the $L_2$ distance are also reported in the appendix.)

**Tangent Space and Jacobian**    The tangent space of the simplex is $T_y \Delta^{K-1} = \{v \in \mathbb{R}^K \mid \mathbf{1}^\top v = 0\}$. The classifier's Jacobian is denoted as $J(x) = \partial y(x)/\partial x \in \mathbb{R}^{K \times d}$, and its regularized pseudoinverse is
$$J_\lambda^\dagger(x) = J(x)^\top \big(J(x)J(x)^\top + \lambda I\big)^{-1}, \qquad \lambda > 0$$
.

**Control Objective**    We aim to design a continuous-time control system $\dot{x}(t) = u(x(t))$ that monotonically decreases $V(x)$ while minimizing the magnitude of the perturbation. In a discrete-step implementation, we ensure a similar property for $x_{t+1} = x_t + \delta x_t$.

## 3 DIRECT OUTPUT CONTROL (DOC)

DOC is a method for repairing misclassifications at inference time based on *minimum-norm output guidance*. Specifically, it defines the distance between the output distribution and a target distribution as a Lyapunov function and pulls back its gradient to the input space via the Jacobian pseudoinverse to compute a minimum-norm perturbation that monotonically decreases the error. This framework uses the Fisher–Rao distance by default but is applicable to other metrics (e.g., $L_2$) as well.

### 3.1 LYAPUNOV FUNCTION AND DESCENT DIRECTION

We define the error function as
$$V(x) = D(y(x), y^\star) \tag{3.1}$$
and its output gradient as $v_y = \nabla_y V(y; y^\star)$. For the Fisher–Rao distance, a closed-form expression can be obtained.

### 3.2 CONTROL LAW

DOC chooses the minimum-norm solution to realize $-v_y$ in the input space as
$$\dot{x} = -J_\lambda^\dagger(x) v_y \tag{3.2}$$
In this case,
$$\dot{V}(x) = -\langle v_y, (JJ_\lambda^\dagger)v_y \rangle \leq 0 \tag{3.3}$$
and the error decreases monotonically.

### 3.3 TOP-K APPROXIMATION

For computational efficiency, we define an active set $S(x)$ consisting of the top $k$ logit components and the target class, and approximate as

$$\dot{x} = -J_S^\top \left(J_S J_S^\top + \lambda I\right)^{-1} v_{y,S} \tag{3.4}$$

based on the partial Jacobian $J_S$. When $v_y$ is contained in the subspace spanned by $J_S$, $V$ similarly decreases monotonically.

### 3.4 FINAL UPDATE RULE

The discrete $N$-step implementation is given by

$$x_{t+1} = x_t - \alpha_t \, J_\lambda^\dagger(x_t) \, v_{y_t}, \qquad y_t = f(x_t) \tag{3.5}$$

where $\alpha_t$ is the step size. This iteration can be seen as an explicit Euler discretization of the continuous-time dynamics

$$\dot{x}(t) = -J_\lambda^\dagger(x(t)) \, v_{y(t)}, \tag{3.6}$$

so that the cumulative displacement over an interval is obtained by integrating the velocity field,

$$\delta x = \int_0^T \dot{x}(t) \, dt = -\int_0^T J_\lambda^\dagger(x(t)) \, v_{y(t)} \, dt. \tag{3.7}$$

Hence, the discrete rule equation 3.5 approximates the integral trajectory that moves $x$ along the pullback of the Lyapunov gradient.

## 4 THEORY: STABILITY, OPTIMALITY, AND IRREPARABILITY

Detailed proofs of the theorems are provided in Appendix D.

### 4.1 T1: LYAPUNOV DECREASE

**Theorem 4.1** (Lyapunov Decrease). *Along the continuous-time system of DOC,*

$$\dot{V}(y) = -\|P_{\mathrm{Im}\, J} \, \nabla_y V(y)\|_2^2 \leq 0 \tag{4.1}$$

*holds.*

**Implication.** DOC always monotonically decreases the error.

### 4.2 T2: MINIMUM-NORM CONTROL

**Theorem 4.2** (Minimum-Norm Control). *Among the inputs that realize $\dot{y} = -P_{\mathrm{Im}\, J}\nabla_y V(y)$,*

$$u_{\mathrm{DOC}} = -J^\dagger \nabla_y V(y) \tag{4.2}$$

*is the unique minimum-norm solution.*

**Implication.** DOC performs repairs with minimal intervention. (Proof in Appendix.)

### 4.3 T3: RELATION TO NATURAL GRADIENT

**Theorem 4.3** (Conditional Natural Gradient Equivalence). *The dynamics of DOC are*

$$\dot{y} = -P_{\mathrm{Im}\, J} \, \nabla_y V(y), \tag{4.3}$$

*which differs from the general natural gradient flow*

$$\dot{y} = -F^+ \nabla_y V(y), \quad F = \mathrm{diag}(y) - yy^\top. \tag{4.4}$$

*However, when $\mathrm{Im}\, J = T_y \Delta^{K-1}$ and the Fisher inner product is used, the two provide the same trajectory up to a time reparameterization.*

**Implication.** DOC and natural gradient are consistent under local conditions. (Proof in Appendix.)

### 4.4 T4: Irreparability Diagnosis (Theoretical Lower Bound)

**Theorem 4.4** (Irreparability Condition). *Under the input constraint $\|\delta x\| \leq \varepsilon$, if*

$$\varepsilon < \frac{\gamma}{\sigma_{\max}(J)} \tag{4.5}$$

*holds, then repair is impossible under a first-order approximation.*

**Implication.** This provides a safe-side irreparability diagnosis, but discriminative power is limited. (Proof in Appendix.)

### 4.5 T5: Decrease Condition for Discretization

**Theorem 4.5** (Monotonic Decrease in Discrete Steps). *For the Euler update*

$$x_{t+1} = x_t - \alpha_t J^{\dagger_\lambda}(x_t) P_t g_t, \tag{4.6}$$

*if the step size $\alpha_t$ is sufficiently small,*

$$V(x_{t+1}) \leq V(x_t) \tag{4.7}$$

*holds.*

**Implication.** Monotonic decrease can be maintained in discrete implementation with proper step size. (Proof in Appendix.)

## 5 Related Work

**Test-Time Adaptation (TTA).** TTA updates model parameters during inference to handle distribution shifts. Tent Wang et al. (2021) minimizes output entropy, EATA Niu et al. (2022) adds confidence-based selection and anti-forgetting, MEMO Zhang et al. (2021) uses augmented views, SAR Mirza et al. (2023) suppresses sharpness, and TTT Sun et al. (2020) employs self-supervised tasks. Extensions to vision‐language models include WATT Wang et al. (2024b) (weight averaging) and TTPA Mao et al. (2023) (prompt adaptation). These methods are effective but rely on parameter updates, with computational cost, forgetting risk, and no guarantees of monotonic repair.

**Purification- and Optimization-based Perturbation Methods.** Another line of work applies controlled input perturbations to restore or defend predictions. PixelDefend projects adversarial inputs back toward the data manifold using a generative prior Song et al. (2018). Adversarial Purification with Score-based Generative Models removes perturbations via score-based energy models Yoon et al. (2021). DiffPure leverages forward‐reverse diffusion to purify inputs Nie et al. (2022). ScoreOpt performs inference-time optimization guided by learned score priors Zhang et al. (2023). DiffHammer revisits diffusion-based purification and exposes evaluation weaknesses under stronger attack protocols Wang et al. (2024a). Nearest-neighbor manifold projection offers a retrieval-based alternative to realign inputs Dubey et al. (2019). Bridge-model purification further steers diffusion trajectories for reliable recovery Li et al. (2024). While effective at input-space correction, these methods target defense or denoising and do not provide guarantees of monotonic repair or irreparability diagnosis.

**Output Calibration and Inference Control.** Post-hoc approaches that regulate the output distribution without modifying model parameters have also been studied. Temperature Scaling (TS) (Guo et al. (2017)) calibrates predicted probabilities with a single temperature parameter, while Dirichlet Calibration (Kull et al. (2019)) extends calibration to multiclass classification. For addressing long-tail distributions, Logit Adjustment (Menon et al. (2021)) provides a unified perspective on post-hoc correction and loss modification. In vision‐language models, methods such as BoostAdapter (Yang et al. (2024)), which enables gradient-free adaptation for CLIP, represent recent progress in TTA and post-hoc control. While these approaches improve probability calibration and mitigate bias, they do not provide a framework guaranteeing stable repair or irreparability diagnosis.

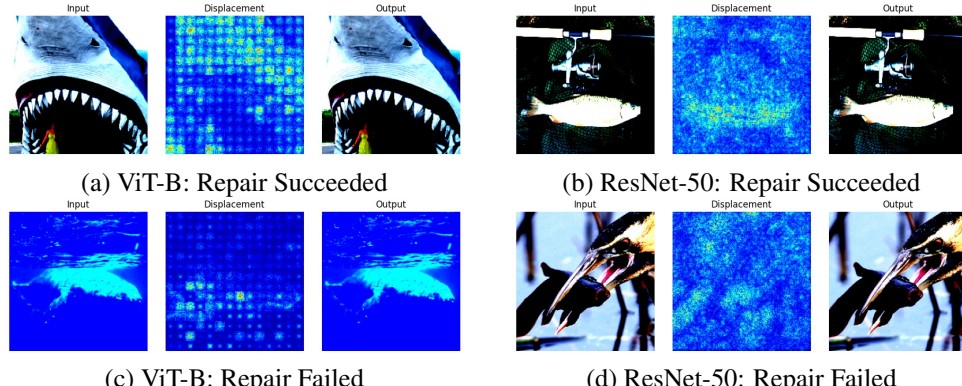

(a) ViT-B: Repair Succeeded        (b) ResNet-50: Repair Succeeded

(c) ViT-B: Repair Failed        (d) ResNet-50: Repair Failed

Figure 2: **Case studies of repair outcomes.** Each block shows Input‑ Displacement‑ Output. ViT-B (left) and ResNet-50 (right) exhibit successful repairs with monotonic Lyapunov decrease (top), and failed repairs with stagnation despite comparable distortions (bottom). The displacements are contrast-enhanced for visualization; in practice the differences are imperceptibly small.

**Position of Our Work.** Direct Output Control (DOC) establishes a new paradigm of inference-time control that addresses the limitations of the above approaches. Like TTA, it performs adaptation during inference, but without parameter updates; like adversarial perturbation methods, it introduces minimal input perturbations, but its objective is stable repair rather than attack. Specifically, DOC defines the Fisher‑ Rao distance on output distributions as a Lyapunov function, pulls back its gradient to the input space via the Jacobian pseudoinverse, and thereby guarantees monotonic error reduction. Furthermore, it derives diagnostic criteria for irreparability based on Jacobian singular values and inter-class margins, explicitly revealing the "limits of repair" absent in prior work. Consequently, DOC is positioned as the first inference-time control framework that simultaneously achieves stability, repair, and diagnosis.

# 6 EXPERIMENTS

## 6.1 SETUP

In this study, we use the ImageNet-1k validation dataset to evaluate both convolutional models (ResNet-50, ResNet-101) and transformer-based models (ViT-B, ViT-L). We extracted misclassified samples from each model and used them as subjects for our repair experiments. The comparison methods used are DOC, PGD, and DeepFool. The evaluation metrics were Top-1/Top-5 repair success rate, distortion metrics ($L_2$, LPIPS), and computational overhead.

## 6.2 EVALUATION PROTOCOL

We conducted three complementary experiments: (i) **Repair Accuracy**: direct comparison of success rate and distortion across methods; (ii) **Diagnosis via $\Delta V$**: ROC analysis using the decrease of the Lyapunov function, evaluating AUC and related metrics; (iii) **Lyapunov Decrease**: trajectory analysis over 100 misclassified samples for ResNet-50 and ViT-B, reporting typical success/failure cases and average curves.

# 7 EXPERIMENTAL RESULTS

## 7.1 REPAIR PERFORMANCE ON IMAGENET-1K

Table 1 shows the trade-off between repair accuracy and distortion. For all ResNet and ViT models, DOC achieved the highest Top-1/Top-5 repair success rates and also had the minimum distortion in both $L_2$ distance and LPIPS. PGD attained moderate success rates but required large perturbations, while DeepFool had the lowest computational overhead but achieved substantially lower repair suc-

cess rates. Figure 3 visualizes the Pareto frontier of repair success rate and distortion, showing that DOC exhibits Pareto superiority over existing methods in both geometric ($L_1$, $L_2$) and perceptual (LPIPS) aspects.

Table 1: **Comparison of DOC, PGD, and DeepFool on ResNet and ViT models.** Best values are shown in bold. The second column shows Clean accuracy (Top-1 / Top-5).

| Model | Clean Acc (Top-1/Top-5) | Method | Top-1 Success | Top-5 Success | $L_2$ Dist. | LPIPS | Overhead (s) |
|---|---|---|---|---|---|---|---|
| ResNet50 | 76.1 / 92.9 | **DOC** | **0.900** | **0.924** | **0.003** | $< 0.001$ | 0.811 |
| | | PGD | 0.692 | 0.837 | 0.842 | 0.351 | 0.248 |
| | | DeepFool | 0.230 | 0.335 | 0.500 | 0.208 | **0.199** |
| ResNet101 | 77.4 / 93.6 | **DOC** | **0.889** | **0.921** | **0.003** | $< 0.001$ | 1.540 |
| | | PGD | 0.684 | 0.848 | 0.848 | 0.351 | 0.479 |
| | | DeepFool | 0.010 | 0.024 | 0.363 | 0.150 | **0.399** |
| ViT-B | 81.1 / 95.7 | **DOC** | **0.859** | 0.913 | **0.005** | $< 0.001$ | 1.089 |
| | | PGD | 0.795 | **0.961** | 0.885 | 0.352 | **0.278** |
| | | DeepFool | 0.239 | 0.884 | 0.885 | 0.352 | 0.099 |
| ViT-L | 84.4 / 97.2 | **DOC** | **0.880** | 0.926 | **0.004** | $< 0.001$ | 2.301 |
| | | PGD | 0.730 | **0.957** | 0.868 | 0.345 | **0.533** |
| | | DeepFool | 0.207 | 0.850 | 0.868 | 0.345 | 0.238 |

## 7.2 REPAIR SUCCESS/FAILURE CLASSIFICATION USING $\Delta V$

Table 2 shows the ROC-based evaluation. DOC achieved an AUC of 0.92 - 0.97 for all models, with Precision/Recall/F1 all above 0.95, indicating high discriminative performance. PGD showed moderate performance, while DeepFool performed significantly worse. Figure 4 shows the ROC curves and the distribution of $\Delta V$. It is confirmed that DOC clearly separates success and failure, whereas the boundary is ambiguous for PGD and DeepFool.

## 7.3 LYAPUNOV FUNCTION DECREASE ANALYSIS

Figure 5 shows the trajectories of the Lyapunov value. In successful cases, DOC showed consistent monotonic decrease and convergence to the target distribution. PGD decreased but converged slowly, while DeepFool tended to decrease initially and then stagnate. The average behavior over 100 misclassified samples also confirmed that DOC exhibits the most stable decrease characteristics.

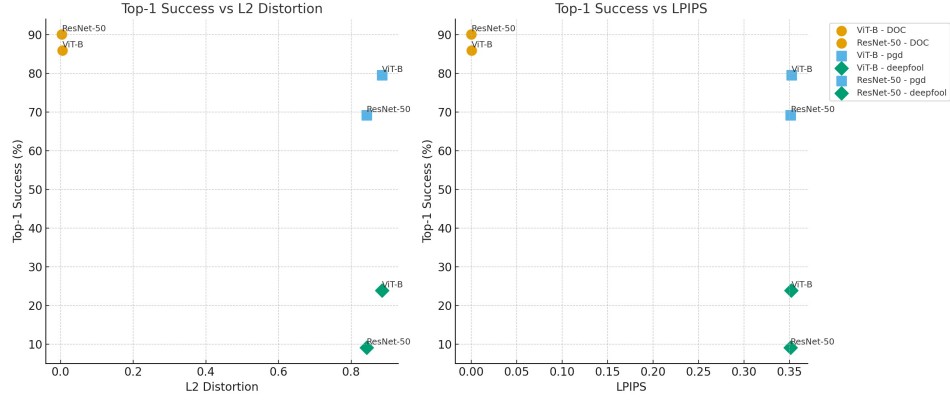

Figure 3: **Pareto plot of repair success rate vs. distortion on ImageNet-1k.** Left: (a) Top-1 Success vs. $L_2$ Distortion. Right: (b) Top-1 Success vs. LPIPS. DOC shows Pareto superiority over existing methods in terms of both geometric and perceptual distortion.

Table 2: **Repair Success/Failure Classification Results using $\Delta V$ (Youden's $J$ statistic).** For all models and metrics, DOC achieved the highest scores. PGD also showed high accuracy, but DOC surpassed it in all cases.

| Model | Method | AUC | Accuracy | Precision | Recall | F1 |
|---|---|---|---|---|---|---|
| ResNet-50 | DOC | **0.97** | **0.98** | **0.99** | **0.99** | **0.99** |
| | PGD | 0.94 | 0.88 | 0.93 | 0.89 | 0.91 |
| | DeepFool | 0.78 | 0.73 | 0.21 | 0.70 | 0.32 |
| ResNet-101 | DOC | **0.97** | **0.99** | **0.99** | **1.00** | **0.99** |
| | PGD | 0.93 | 0.87 | 0.92 | 0.89 | 0.91 |
| | DeepFool | 0.81 | 0.75 | 0.03 | 0.73 | 0.06 |
| ViT-B | DOC | **0.96** | **0.97** | **0.97** | **0.99** | **0.98** |
| | PGD | 0.91 | 0.82 | 0.95 | 0.82 | 0.88 |
| | DeepFool | 0.72 | 0.66 | 0.38 | 0.65 | 0.48 |
| ViT-L | DOC | **0.92** | **0.93** | **0.95** | **0.97** | **0.96** |
| | PGD | 0.85 | 0.75 | 0.93 | 0.71 | 0.80 |
| | DeepFool | 0.60 | 0.73 | 0.34 | 0.31 | 0.32 |

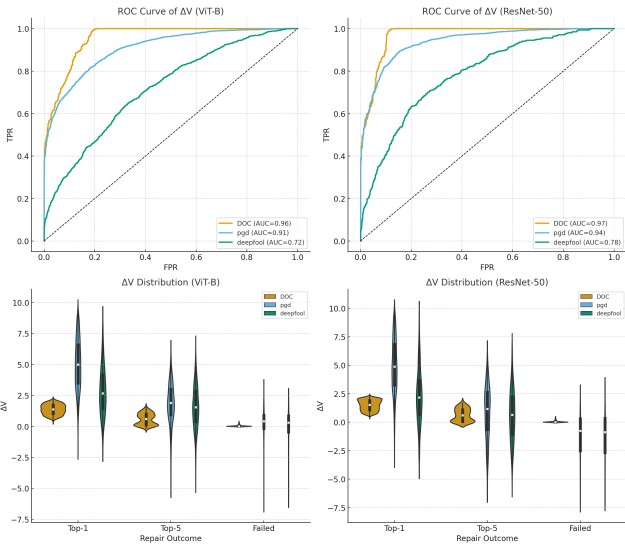

Figure 4: **Comparison of ROC curves and $\Delta V$ distributions for ViT-B and ResNet-50.** DOC consistently achieves higher AUC values than PGD and DeepFool. The violin plots show that for DOC, failed repairs yield $\Delta V$ values concentrated near zero, while successful Top-1 and Top-5 repairs exhibit clearly larger $\Delta V$, indicating strong discriminative power.

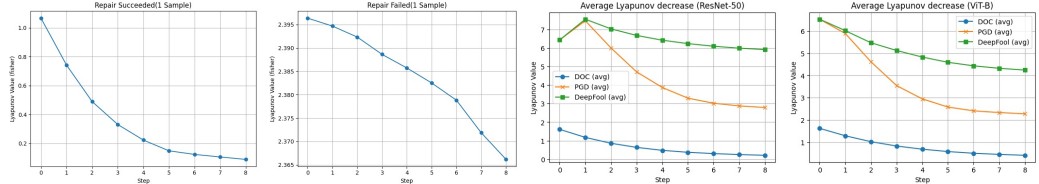

Figure 5: **Trajectories of Lyapunov values**. (Far left) Transition of Lyapunov values for successful repair cases, (Second from left) Transition of Lyapunov values for failed repair cases, (Second from right) Average decrease of Lyapunov values for DOC, PGD, and DeepFool on ResNet-50, (Far right) Average decrease of Lyapunov values for DOC, PGD, and DeepFool on ViT-B.

## 8    DISCUSSION

**Failure Modes and Diagnosis.**    Failures may occur when the Jacobian rank collapses, when the margin $\gamma$ relative to $\sigma_{\max}(J)$ is too large, or when the Top-$k$ set is insufficient. Theorem 4.4 provides an absolute lower bound but is costly, while the empirical decrease $\Delta V$ is efficient but lacks guarantees. Using both offers a dual system: theory for safety, $\Delta V$ for practicality. In practice, Theorem 4.4 can act as an absolute but costly safeguard, while $\boxtimes$ V serves as an efficient proxy. Their combination balances theoretical guarantees with practical usability.

**Practical Use.**    DOC can be added without retraining. Repair can be triggered and terminated by monitoring $\Delta V$, with irreparable cases flagged for auditing. Overhead is dominated by pseudoinverse approximation but can be reduced with Top-$k$ or randomized methods. Since labels must be specified, repair requires either human-in-the-loop supervision or auxiliary modules that obtain external information.In practice, further efficiency can be achieved with randomized low-rank solvers or Hutch++ estimators, making DOC feasible even for large-scale deployment.

**Theory‑ Practice Gap.**    Safety guarantees are expensive, heuristics are efficient. Approximate stability criteria (e.g., Lipschitz bounds or trust regions) may make $\Delta V$ monitoring more reliable without full spectral analysis.

**Comparison.**    TTA updates parameters but lacks stability guarantees. Attack-based methods cross boundaries but lack diagnostics. DOC instead provides inference-time control with monotone decrease, minimal intervention, and diagnosability.

## 9    LIMITATIONS AND SOCIETAL IMPACT

**Technical Limitations.**    Irreparability checks remain sufficient but not necessary. Extreme imbalance or multi-label settings may break the geometry. Pseudoinverse approximations and active set choices introduce variability. Non-classification tasks require adapting the Lyapunov definition. Label specification remains a structural limitation, motivating future work on semi-supervised or automated repair mechanisms that can infer target labels from auxiliary information.

**Reproducibility.**    Results depend on preprocessing, metrics, seeds, and approximation details. Publishing code, logs, and definitions of $\Delta V$ and stopping rules is essential.

**Societal Impact.**    DOC can improve safety in high-risk domains by repairing errors and flagging irreparable cases without altering model weights. Risks include misuse for adversarial editing, bias reinforcement, and higher compute cost. Mitigation requires audit logging, fairness checks, and clear policies for escalation to human oversight.To further reduce misuse risks, repair logs and $\boxtimes$ V-based audit trails can be mandated as part of accountability frameworks.

## 10    CONCLUSION

This paper proposed *Direct Output Control*, which directly controls the output distribution without modifying the trained model. By using the distance $V$ as a Lyapunov function and the Jacobian pseudoinverse, we guaranteed **monotonic decrease** and derived a **minimum-norm** input perturbation. Furthermore, we provided an **irreparability diagnosis** based on Jacobian properties and $\Delta V$. On ResNet and ViT models with ImageNet-1k, DOC demonstrated **Pareto superiority** over TTA and PGD/DeepFool in terms of both repair success rate and distortion, and $\Delta V$ was able to distinguish success from failure with a high AUC. DOC represents a framework that shifts the paradigm from attack optimization to control optimization, serving as a practical foundation that integrates safe inference-time repair and diagnosis. Future work will proceed towards extending metrics, integrating CLF‑ CBF, expanding to multi-modal applications, and learning operational policies driven by $\Delta V$.Future work will also explore scalable approximations and audit-based deployment frameworks to ensure both efficiency and accountability.

## REGARDING THE USE OF LLMs

In the preparation of this study, a large language model (LLM) was used for writing assistance, searching for related research, and as an aid during the initial brainstorming phase of the research. In writing, it was utilized to refine grammar and clarity of expression. For related research, it was used to efficiently grasp literature and relevant fields, but the final adoption and citation of sources were all verified by the authors. Additionally, in the early stages of research, it was used as a discussion aid to consider alternative formulations and approaches. It should be noted that the final responsibility for all scientific claims, experiments, and conclusions in the paper rests entirely with the authors, and the output of the LLM was always verified by a human before adoption.

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
