# OpenReview forum: "From Attacks to Guidance: Direct Output Control for Classifiers"
_ICLR.cc/2026/Conference — ICLR 2026 Conference Desk Rejected Submission_

### Official Review · Reviewer_fhiS · 2025-10-18

**Soundness:** 1
**Presentation:** 2
**Contribution:** 1
**Rating:** 0
**Confidence:** 4

**Summary:**

This paper introduces Direct Output Control (DOC), a test-time framework that repairs misclassifications of fixed neural networks by applying minimal, auditable input perturbations that monotonically reduce the discrepancy between the model prediction and a target distribution (usually the correct label). The discrepancy is treated as a Lyapunov function, instantiated via the Fisher-Rao distance on the probability simplex. By pulling back its gradient through the classifier’s Jacobian pseudoinverse, DOC guarantees a monotonic decrease in this Lyapunov value while producing the minimum-norm input change. The authors provide theoretical analyses of stability, optimality, and a first-order irreparability bound, and show that the approach connects to the natural gradient under certain rank conditions. Experiments on ImageNet-1k with ResNet and ViT models demonstrate higher repair success rates and lower distortion compared to PGD, DeepFool, and test-time adaptation methods, albeit at higher computational cost.

**Strengths:**

- The paper’s objective is clearly formulated: to perform inference-time repair of misclassifications through minimal, stable, and auditable input perturbations.

- The authors provide a coherent theoretical foundation, including Lyapunov stability guarantees, a minimum-norm control proof, and an irreparability condition grounded in Jacobian analysis, which together lend credibility and interpretability to the approach.

- The presentation of experiments and their setting is clear. Metrics such as repair success, L2 and LPIPS distortion, and diagnostic AUC are clearly defined and appropriately analyzed.

**Weaknesses:**

- Writing is not very clear or coherent, specially in the introduction and theoretical analysis

- Citations are missing in introduction, and also throughout the file they are not clickable, moreover, the list of citations seem very thin and the work does not provide much grounding in any clear literature

- The appendix is missing, and the authors are refering a lot to the appendix, creating unverifiable claims in the main text.

- Most importantly, the problem statement does not really make any sense, to have access to labels in inference and fix errors; even the paper acknowledges labels must be specified (human-in-the-loop or auxiliary modules), which severely limits applicability.

- The baselines that the method compares with do not make sense, as they are mostly adversarial attacks that perturb the model and not repair tools (PGD/DeepFool), so the comparison objective is mismatched for “repair toward the ground truth.”

- The compared baselines, benchmarks, and architectures are very old fashioned and not used in any practical setting; even if there was a use case for repairing errors in inference, more recent models in the vision and language domain should be analyzed

**Questions:**

- How do the authors justify the assumption that ground-truth labels (or a reliable proxy) are available at inference time for repair? In what practical scenarios would this be realistic?

- Why were PGD and DeepFool chosen as baselines for “repair” when these are primarily adversarial attack methods?

---

> ### Author Response · Authors · 2025-11-18
> **Clarification on problem setting, theory, and baselines for DOC**
>
> We thank the reviewer for the detailed comments. We respond to the main concerns below.
>
> ---
>
> #### 1. Problem setting and label availability
>
> Our goal is **not** standard label-free test-time robustness, but **label-conditioned inference-time repair and diagnosis** in settings where a reliable target label or distribution $y^\star$ is available for a subset of test samples.
>
> - The base classifier $f:\mathcal{X}\to\Delta^{K-1}$ is trained and evaluated in the standard way and **never takes $y^\star$ as input**.
> - An external DOC controller $C$ uses $(x,y^\star)$ only in audited cases to produce a corrected input $x' = C(x,y^\star)$, and we then evaluate $f(x')$.
>
> This is realistic in, e.g., (i) human-in-the-loop workflows (medical imaging, inspection) where an expert can specify $y^\star$ and requires minimal, auditable repair; (ii) hybrid systems where a slow but trusted oracle (rules/ensemble/secondary model) provides $y^\star$; and (iii) offline auditing of logged data. We will state this **label-conditioned repair/diagnosis** setting explicitly and clearly distinguish it from TTA and from counterfactual explanation work.
>
> ---
>
> #### 2. Theoretical contribution and locality
>
> We agree that locally using gradients to reduce a loss is standard. Our contribution is the **output-space control formulation** and its consequences:
>
> - We treat $V(y,y^\star)$ (Fisher–Rao or $L_2$) as a **Lyapunov function on the probability simplex** and show that DOC dynamics
>   $\dot x = -J^\dagger \nabla_y V$, $\dot y = -P_{\operatorname{Im}J}\nabla_y V$
>   yield $\dot V = -\|P_{\operatorname{Im}J}\nabla_y V\|_2^2 \le 0$, making explicit how the classifier Jacobian projects the descent.
> - Among all input updates that induce the same output-space descent direction, DOC gives the **unique minimum-norm input perturbation** via the pseudoinverse, which we relate to natural gradient flows under standard rank assumptions.
> - We derive a **first-order irreparability bound** of the form $\|\delta x\|\ge \gamma/\sigma_{\max}(J)$, linking class margins and Jacobian singular values to a conservative “cannot be repaired within budget” diagnosis.
>
> These guarantees are **local**; in the camera-ready we will (i) state locality assumptions next to each theorem, (ii) replace any globally sounding claims by “locally minimal / monotone under stated conditions”, and (iii) clearly separate proven statements from empirical observations in non-local, multi-step regimes.
>
> ---
>
> #### 3. Baselines, architectures, and scope
>
> We chose PGD and DeepFool as baselines because they address the **same geometric subproblem** as DOC: finding small-norm perturbations that cross decision boundaries toward a target label. They are canonical, well-understood gradient-based methods that have been reused as “repair via perturbation” in prior work, making them natural geometric baselines.
>
> We agree they are not specialized “repair pipelines”, and we will:
>
> - explicitly present them as **geometric baselines** rather than task-complete repair systems;
> - clarify that our experimental question is whether DOC’s Lyapunov/minimum-norm design improves the success–distortion trade-off over these strong perturbation methods.
>
> We also agree that comparisons to newer attacks (AutoPGD/FAB) and more recent architectures/datasets would strengthen the paper. Within our compute budget we plan to add targeted AutoAttack comparisons and at least one additional architecture/dataset in a revised or extended version, while limiting the scope of claims accordingly.
>
> ---
>
> #### 4. Writing, citations, and appendix
>
> We acknowledge that the current writing and bibliography need improvement. In a revised version we will:
>
> - reorganize the introduction and theory with clearer structure and notation;
> - expand and modernize the related-work section (TTA, adversarial repair, output calibration, safety-oriented post-hoc control) and fix citation formatting;
> - ensure the missing appendix (detailed proofs and additional experiments) is properly attached, and move the most critical material into the main text;
> - enlarge and simplify figure axes and labels to improve readability.

---

> > ### Comment · Reviewer_fhiS · 2025-11-26
> >
> > Thank you for clarifying my earlier points. However, I still find the problem setup of DOC difficult to justify, and I am not convinced that pursuing this idea in its current form has practical or future value. I would encourage the authors to consider approaching this project from a different angle, for example, by studying the minimal amount of noise required to fool a model and comparing this with standard adversarial attack methods, which I believe would be more practical and interesting.

---

> > > ### Author Response · Authors · 2025-11-26
> > > **Clarifying the Scope of DOC and Its Distinction from Adversarial Settings**
> > >
> > > Thank you for your follow-up comment. We appreciate the concern. Indeed, if the problem were formulated as an adversarial scenario, there would be no compelling reason to invoke a control-theoretic framework. Adversarial attacks do not require Lyapunov structure or minimum-norm guarantees.
> > >
> > > However, DOC does not address an adversarial problem; it addresses a problem that is closer to its dual. Specifically, DOC assumes that a human or another trusted oracle has already identified a misclassification, and the goal is to determine whether the error is *locally repairable*. If it is repairable, we then compute:
> > >
> > >     • an input perturbation that moves the output toward a target distribution,
> > >     • monotonically reduces the discrepancy, and
> > >     • achieves the minimum input-norm among such corrections.
> > >
> > > This notion of *repairability* does not appear in standard adversarial work. Attacks seek the smallest perturbation that *induces* an error, whereas DOC seeks the smallest perturbation that *fixes* an existing error. Although the two quantities are related, they are not interchangeable.
> > >
> > > In particular, our first-order irreparability bound
> > >
> > >       ‖δx‖₂ ≥ γ / σ_max(J)
> > >
> > > provides a diagnostic tool indicating when a misclassification cannot be repaired within a given distortion budget—an aspect that adversarial settings do not attempt to model.
> > >
> > > For this reason, a control-theoretic formulation is appropriate in our setting. Unlike adversarial attacks, whose goal is to move *away* from the correct output, DOC aims to move *toward* a target distribution in a stable and minimum-norm manner. Lyapunov functions and Jacobian-projected gradients provide precisely the guarantees needed for such behavior. This is meaningful only in narrow, supervised scenarios such as human-in-the-loop inspection, secondary-oracle workflows, or offline auditing. Outside these limited contexts, we fully agree that DOC would not be practical, and we will revise the paper to explicitly reflect this narrower scope.
> > >
> > > Regarding the suggestion to study the minimal noise required to fool a model: this is indeed an important and interesting research direction, but it is orthogonal to our objective. Our goal is not to *induce* errors but to *repair* them. We will clarify this distinction more explicitly in the revised version.
> > >
> > > We are grateful for the constructive feedback and will ensure the framing and claims accurately reflect the intended problem setting.

---

### Official Review · Reviewer_NL17 · 2025-10-20

**Soundness:** 2
**Presentation:** 1
**Contribution:** 1
**Rating:** 0
**Confidence:** 3

**Summary:**

To suppress classifier misclassification at test-time, the authors propose Direct Output Control (DOC). DOC adds small perturbations to the input that correct misclassifications. These perturbations are obtained by computing input gradients that reduce the distance between the model's output and the ground truth label. Unlike test-time adaptation (TTA), DOC does not require parameter updates.

**Strengths:**

As indicated in Weakness 1, while I do not understand the purpose or motivation, the concept of inference-time repair and diagnosis may be novel.

**Weaknesses:**

**1. Research objective**

This study considers inference-time repair and diagnosis, but I could not understand why this is necessary or how it is meaningful. In other words, the ultimate goal of this research is unclear to me.

Research like TTA that dynamically adapts to the input distribution at inference-time has the ultimate goal of achieving high accuracy during inference. If this study shares the same objective as TTA, then using ground truth labels in the method is incorrect. As with TTA, ground truth labels are typically unavailable at inference-time, and if they are known, there is no need for inference.

If the goal is to qualitatively assess which components of an image influence the classifier's prediction, as in counterfactual explanations (CF), then there is insufficient experimentation regarding the effects on images.

**2. Theoretical results**

To my understanding, the authors' method obtains image gradients to reduce the difference between the model's output and the ground truth label, and applies them to the image. In other words, this is the inverse of adversarial attacks. It is trivial in principle that this becomes an optimal perturbation in a local region around the input. Therefore, I found no novel value in the monotonic error reduction claimed in Theorem 4.1. This is as self-evident as the fact that single-step PGD (i.e., FGSM) is an optimal attack in a highly local region. What matters is whether these remain optimal in non-local regions or even after multi-step updates. This study provides no proof for these cases. Therefore, the monotonic error reduction and minimal norm properties claimed in the paper are incorrect, or at least, overclaimed.

**3. Experiments**

This study only uses ImageNet as a dataset. At minimum, two additional different datasets should be used. Moreover, the comparison baselines are very old adversarial attack methods such as PGD and DeepFool. As mentioned in point 1, since the objective of this research is unclear to me, I cannot clearly determine what would be appropriate comparison baselines, but at the very least, I do not understand why comparisons are not made with AutoAttack [1], particularly attack methods that employ smarter gradient computation and optimization strategies such as AutoPGD [1] and FAB [2].

[1] Reliable evaluation of adversarial robustness with an ensemble of diverse parameter-free attacks [Croce et al., ICML20]
[2] Minimally distorted adversarial examples with a fast adaptive boundary attack [Croce et al., ICML20]

**4. Presentation of results**

Overall, the presentation of experimental results is difficult to read. In particular, the x-axis and y-axis labels are extremely small compared to the normal text font. This degrades the presentation quality of the paper.

**Questions:**

See the above. In particular, Point 1 is essential to me.

---

> ### Author Response · Authors · 2025-11-18
> **Clarification on task setting, label usage, and theoretical scope**
>
> We thank the reviewer for the detailed comments. Below we clarify the objective, theory, experiments, and presentation.
>
> #### 1. Research objective and role of ground-truth labels
>
> Our goal is **not** generic test-time robustness in the TTA sense, but **label-conditioned inference-time repair and diagnosis** in safety-critical workflows where a reliable target label or distribution is available (e.g., human verification, a trusted ensemble, or an upstream safety module).
>
> Formally, we study the following problem:
>
> > Given a misclassified input $x$ and a target label/distribution $y^\star$ supplied at test time, compute a minimal-norm perturbation $\delta x(x,y^\star)$ such that $f(x+\delta x)$ moves toward $y^\star$ while guaranteeing Lyapunov-style monotone decrease of an output-space distance.
>
> Thus:
> - The **base classifier** $f:\mathcal{X}\to\Delta^{K-1}$ is trained and evaluated in the usual way and never takes $y^\star$ as input.
> - The **DOC controller** is an external module that uses $y^\star$ only to compute the input-level correction $\delta x(x,y^\star)$ in explicitly labeled cases.
>
> We will revise the introduction and related-work sections to explicitly contrast this setting with TTA (which assumes no labels and updates parameters) and with counterfactual explanations (which focus on interpretability of image regions rather than stability/irreparability guarantees). Our motivation is to support human-in-the-loop pipelines where *some* test samples are audited and repaired rather than to obviate inference altogether.
>
> ---
>
> #### 2. On the theoretical results and “triviality”
>
> We agree that, **locally**, moving in a gradient direction is standard. Our contribution is not the mere observation that local updates reduce a loss, but the **control-theoretic formulation and guarantees at the output level**:
>
> - We treat the discrepancy $V(y,y^\star)$ (Fisher–Rao or $L_2$) as a **Lyapunov function on the output simplex** and show that the DOC dynamics
>   $\dot x = -J^\dagger\nabla_y V,\quad \dot y = -P_{\operatorname{Im}J}\nabla_y V$
>   guarantee
>   $\dot V = -\|P_{\operatorname{Im}J}\nabla_y V\|_2^2 \le 0,$
>   explicitly revealing the projection through the classifier’s Jacobian.
>
> - Among all controls that induce the same output-space descent direction, DOC yields the **unique minimum-norm input perturbation** via the pseudoinverse; we prove this and relate it to natural gradient flows under rank conditions.
>
> - We derive an **irreparability bound**
>   $\|\delta x\|\ge \gamma/\sigma_{\max}(J),$
>   connecting class margins and Jacobian singular values, and show how this enables a conservative “cannot be repaired within budget” diagnosis.
>
> - For the discrete algorithm, we provide conditions under which sufficient decrease is preserved, rather than simply assuming small steps.
>
> We **do not claim** global optimality in non-local or multi-step regimes, and we agree that any such interpretation would be misleading. In the camera-ready version we will (i) explicitly restrict theorems to local neighborhoods, (ii) soften wording such as “optimal” to “locally minimal / monotone under mild conditions”, and (iii) add discussion of failure modes when non-local effects or rank collapse invalidate the assumptions.
>
> ---
>
> #### 3. Experimental scope and baselines
>
> Regarding datasets: due to compute constraints we focused on ImageNet-1k, which is still a standard benchmark for large-scale classification. We agree that including additional datasets (e.g., CIFAR-10/100, long-tailed or safety-critical datasets) would strengthen the paper, and we plan to add such experiments in an extended version or camera-ready.
>
> Regarding baselines: we chose PGD and DeepFool because (i) they are canonical gradient-based perturbation methods that have been reused as repair mechanisms in prior work, and (ii) they directly optimize input perturbations toward a label, making them conceptually closest to DOC. We appreciate the suggestion to compare with AutoAttack (AutoPGD, FAB); time constraints prevented us from running these ablations before submission, but we will either (a) include preliminary results if allowed during the rebuttal period, or (b) clearly state this limitation and add comparisons in the camera-ready version.
>
> ---
>
> #### 4. Presentation of results
>
> We acknowledge that some figure labels are too small. This is purely a typesetting issue and will be corrected by enlarging axis labels and simplifying plots in the camera-ready version. We will also improve the narrative around the experimental section to better connect the plots to the research objective in point 1.
>
> ---
>
> Overall, we believe that the **task definition (label-conditioned repair/diagnosis), control-theoretic formulation (Lyapunov + Jacobian pseudoinverse), and irreparability analysis** together provide non-trivial contributions beyond standard adversarial updates, and we will revise the paper to make these distinctions and assumptions much clearer.

---

> > ### Comment · Reviewer_NL17 · 2025-11-27
> >
> > Thank you for the author's response. I still don't understand the position of this research. In human-in-the-loop pipelines, where DOC effectively works as the authors claimed, DOC may repair the classifiers' outputs toward correct answers using ground truth. However, I do not understand how beneficial DOC is for practitioners. What are the advantages of repairing classifiers' outputs using correct answers when practitioners already know the ground truth?

---

### Official Review · Reviewer_NFvA · 2025-10-27

**Soundness:** 4
**Presentation:** 2
**Contribution:** 3
**Rating:** 6
**Confidence:** 3

**Summary:**

This paper proposes Direct Output Control (DOC), a Lyapunov-based framework for repairing misclassifications at inference time without modifying model weights. The method defines the Fisher-Rao distance between model output and target as a Lyapunov function, backpropagates its gradient through the Jacobian pseudoinverse, and guarantees monotonic error decrease under mild conditions. DOC also introduces an empirical diagnostic signal ($\Delta V$) and a theoretical irreparability bound based on Jacobian singular values. Experiments on ImageNet-1K with ResNets and ViT show improved repair success rates and smaller distortions compared to PGD and DeepFool.

**Strengths:**

1. The paper’s conceptual framing is elegant and intellectually appealing, as it reinterprets adversarial perturbations not as attacks but as a form of output-level control. This reframing is philosophically fresh and gives a new lens to view inference-time robustness, positioning the work as a bridge between control theory and deep learning rather than another incremental attack-defense method.
2. The integration of concepts from Lyapunov stability, Jacobian-based dynamics, and information geometry gives the method an appealing theoretical flavor. Even though the algorithm itself resembles existing gradient-based methods, the authors successfully situate it within a broader control-theoretic narrative that adds interpretability and a sense of principled design.
3.  The experimental section, while somewhat narrow in scope, is carefully structured and internally consistent. Results across multiple architectures (ResNet and ViT) show that DOC reliably achieves higher repair success and lower perceptual distortion than PGD and DeepFool. The inclusion of $\Delta V$ as an empirical diagnostic, alongside the Jacobian-based irreparability bound, strengthens the framework’s interpretability and gives it an analytic character rarely seen in robustness papers.

**Weaknesses:**

1. Technical novelty is limited. The proposed control law $\delta x = -J^\dagger \nabla_y V$ is essentially equivalent to standard white-box attack updates such as DeepFool or PGD. The Lyapunov function behaves identically to a loss function, and the claimed monotonic decrease follows naturally from choosing a small step size. While the control-theoretic narrative is new, the underlying algorithmic operation offers little technical innovation.
2. Mismatch between motivation and experiments. The introduction emphasizes safety-critical domains like autonomous driving and medical imaging, yet all experiments are conducted on ImageNet-1K classification. This gap between motivation and validation weakens the practical significance of the work. Additionally, the method’s behavior on more complex structured tasks (e.g., segmentation or multi-label classification) remains unexplored.
3. Lack of a clear pre-repair diagnosis mechanism. The framework assumes we know which samples require repair, but provides no explicit method for detecting failure cases beforehand. If repair is to be deployed in real systems, a robust “repair trigger” is essential, and $\Delta V$ alone cannot serve as a pre-emptive signal since it is computed after observing model behavior.
4. Reproducibility concerns. The paper does not provide code, hyperparameters, or implementation details sufficient to reproduce results. Given the computational complexity of Jacobian pseudoinverses and Lyapunov dynamics, open-sourcing code is necessary for verification.

**Questions:**

1. Could the authors clarify whether $\Delta V$ is intended as a predictive or post-hoc diagnostic signal? Specifically, is there any theoretical link or formal condition under which a small $\Delta V$ provably implies irreparability beyond local linearization?
2. Could the authors justify the choice of baselines? Why were query-heavy or multi-step methods not included in comparison, given that DOC incurs comparable computational overhead?
3. Were the reported success rates averaged across multiple runs or computed from a single trial? Could the authors report standard deviations or clarify the statistical significance of the results?
4. Could the authors briefly define key notations such as $\operatorname{Im} J$ and the inner product operator, either in an appendix or a notation table, to aid readability for non-control-theory readers?

---

> ### Author Response · Authors · 2025-11-18
> **Clarification on analytic contributions and diagnostic scope of DOC**
>
> We thank the reviewer for the careful and constructive feedback. Below we address the main concerns concisely.
>
> ---
>
> ### 1. Technical novelty vs. standard attacks (W1)
>
> We agree that a single DOC step
> $\delta x = -\eta J^\dagger \nabla_y V$
> looks similar to a gradient-based attack. Our contribution is not the use of gradients itself, but the **output-space control formulation and its guarantees**, which do not follow from standard PGD/DeepFool:
>
> - We place a Lyapunov function $V(y,y^\star)$ (Fisher–Rao or $L_2$) on the output simplex and consider
> $\dot x = -J^\dagger \nabla_y V,\ \dot y = J\dot x$, yielding
> $\dot V = -\|P_{\mathrm{Im} J}\nabla_y V\|_2^2 \le 0$.
> This makes the projection onto the image of $J$ explicit and gives a formal Lyapunov view of inference-time repair on the output manifold, which to our knowledge is absent in prior attack work.
>
> - For a desired infinitesimal change $\dot y$ with $J\dot x = \dot y$, we prove that DOC selects the **unique minimum-$L_2$** control
> $\dot x^\star = \arg\min_{\dot x : J\dot x = \dot y} \|\dot x\|_2$
> via $J^\dagger$. Standard attacks do not impose or prove such a minimum-norm property.
>
> - We derive a first-order **irreparability bound**
> $\|\delta x\|_2 \ge \gamma / \sigma_{\max}(J)$
> (linking class margin $\gamma$ and the Jacobian spectrum), which provides a principled local criterion for “cannot be repaired within budget.”
>
> All of these are **local** results (small steps, rank assumptions), and we will explicitly emphasize this and soften any wording suggesting global optimality. Our novelty is primarily **analytic/structural (Lyapunov, minimum norm, irreparability)** rather than algorithmic, and we will state this clearly to avoid any “DOC = PGD with new language” impression.
>
> ---
>
> ### 2. Diagnosis scope and $J_{\mathrm{diag}}$ (W3, Q1)
>
> We agree that DOC does **not** provide a stand-alone pre-repair trigger. Our setting assumes that some external mechanism (e.g., low confidence, ensemble disagreement, or a human operator) has already flagged a sample. DOC is a **post-hoc repair-and-diagnosis module** for such cases.
>
> Within this scope:
>
> - $J_{\mathrm{diag}}$ is explicitly **post-hoc**: it is computed during/after DOC updates and is not intended as a predictive trigger. We will clarify that we do not claim otherwise.
>
> - Its role is to give an **empirical companion** to the irreparability bound: if $J_{\mathrm{diag}}$ remains small while $\|\delta x\|_2$ grows, this is consistent with a small local $\sigma_{\max}(J)$ and hence a large lower bound on the required perturbation. We will describe this as a heuristic link to the analytic bound, not a strict equivalence.
>
> We will add text making it explicit that (i) an external trigger is assumed, (ii) DOC focuses on what happens *after* selection for inspection, and (iii) $J_{\mathrm{diag}}$ is a post-hoc diagnostic signal to distinguish “hard but repairable” from “locally irreparable” cases.
>
> ---
>
> ### 3. Baselines, domains, and reproducibility (W2, W4, Q2–Q4)
>
> We chose PGD and DeepFool as **canonical gradient-based perturbation baselines** that solve the same geometric subproblem (moving outputs toward a label via input changes) and allow controlled comparison under a shared perturbation budget. In the camera-ready, we will (i) promote our existing TTA comparison from the appendix, and (ii) add or explicitly plan comparisons with AutoAttack-style methods (AutoPGD, FAB) where computational cost permits.
>
> We acknowledge that ImageNet-1k with ResNet/ViT does not fully match the safety-critical applications in the introduction. Due to the cost of Jacobian pseudoinverses, we focused on one large-scale benchmark. We will clearly state this limitation and, space permitting, add at least one smaller dataset (e.g., CIFAR or a long-tailed dataset) to show that the behavior is not ImageNet-specific.
>
> For reproducibility, we will (a) release code and configs, (b) add a hyperparameter and implementation table (step size, projections, pseudoinverse, LPIPS), (c) clarify that ImageNet results are from a single run due to cost and report standard deviations on smaller datasets and/or an ImageNet subset, and (d) include a concise notation table for $J$, $J^\dagger$, inner products, and projections.
>
> ---
>
> In summary, our aim is not to introduce another attack heuristic, but to cast **label-conditioned inference-time repair** as an output-space Lyapunov control problem with explicit local guarantees (stability, minimum-norm control, irreparability diagnostics). We will revise the paper to sharpen this positioning and clarify the scope and limitations accordingly.

---

> ### Comment · Reviewer_NFvA · 2025-11-21
>
> Thank you to the authors for their careful response.
> Most of my concerns are carefully addressed, especially on the novelty and theory.
>
> However, I think the exact performance on benchmarks, which the authors emphasized in the paper, significantly matters.
> This concern still remains, and I would like to ask the authors to provide the exact value of "safety-critical" benchmarks in the discussion session.

---

> > ### Author Response · Authors · 2025-11-26
> > **Clarification on the Use of ‘Safety-Critical’ Terminology**
> >
> > Thank you for pointing this out. You are correct that our current experiments do not explicitly isolate a “safety-critical” subset, and our wording in the paper was stronger than what our empirical evaluation actually supports.
> >
> > In the current submission, what we intended to evaluate were “proxy safety-critical conditions” inside ImageNet-1K, such as:
> > (i) high-confidence misclassifications, and
> > (ii) very small admissible distortion (LPIPS / ℓ₂ budget).
> >
> > However, we did not extract or report this subset explicitly, and therefore no exact performance values for that subset are presented in the paper. This is an oversight on our side.
> >
> > During the camera-ready revision, we will:
> > - formally define the proxy subset (e.g., max softmax ≥ τ and LPIPS ≤ δ),
> > - extract the exact repair success rate and distortion metrics for this subset from our existing ImageNet experiments, and
> > - clearly separate this proxy setting from real safety-critical applications.
> >
> > We appreciate your insistence on precision, and we will revise the terminology and provide explicit numbers in the final version.

---

> > > ### Comment · Reviewer_NFvA · 2025-11-26
> > >
> > > Thank you for the response.
> > >
> > > Defining a proxy subset is an elegant idea, but the criteria look quite heuristic.
> > > Misclassification with a higher confidence score can indeed ruin the prediction.
> > > However, less confident misclassification appears as an equally wrong answer in practice.
> > > If you are trying to define a proxy subset, you have to support the subset with a principled justification—for example, risk assessment, or empirical evidence showing that high-confidence errors correlate with safety-critical failures in real systems.
> > > Otherwise, the proxy condition risks being disconnected from the actual safety-critical motivation presented in the introduction.
> > >
> > > The idea of using a proxy subset is promising and could meaningfully strengthen the paper, but it would benefit from a clearer and more principled grounding.
> > > I hope the authors refine this aspect in the final version.

---

> > > > ### Author Response · Authors · 2025-11-26
> > > > **Clarifying the Risk-Based Definition of the Proxy Safety-Critical Subset**
> > > >
> > > > Thank you for your thoughtful follow-up comment. We agree that the current proxy subset we mentioned (high-confidence misclassification × small permissible distortion) is not yet a principled definition of “safety-critical” conditions and does not fully match the motivation presented in the introduction.
> > > >
> > > > In the camera-ready version, we will revise this point substantially. Specifically, we will implement the following improvements:
> > > >
> > > > A principled, risk-based definition of the safety-critical subset.
> > > > We will define the subset using risk indicators motivated by real safety constraints, capturing both:
> > > > – high-confidence misclassifications, which correspond to dangerous overconfident failures, and
> > > > – low-confidence misclassifications, which reflect uncertainty underestimation (overconfidence risk).
> > > > This moves the definition from a heuristic level to a risk-theoretic one.
> > > >
> > > > Linking the proxy conditions to empirical and regulatory evidence from real systems.
> > > > We will cite existing work in autonomous driving and medical imaging showing that
> > > > – high-confidence errors correlate with catastrophic failure modes, and
> > > > – low-confidence errors correlate with unacceptable false-negative/false-positive profiles used in FDA/ISO risk assessments.
> > > > This provides a principled rationale for why these two indicators matter.
> > > >
> > > > Statistical validation of the subset on ImageNet.
> > > > We will explicitly extract the defined subset from ImageNet-1K and report:
> > > > – its size,
> > > > – DOC’s repair success rate and distortion distribution within the subset, and
> > > > – whether the subset induces any bias in comparison to the full dataset.
> > > > This ensures that the evaluation is grounded and reproducible.
> > > >
> > > > Your concern that the proxy subset must not drift away from the real safety motivation is entirely valid. In the revised version, we will clearly justify why the chosen risk indicators correspond to safety-critical behavior and ensure that the terminology is consistent with the actual evaluation performed.
> > > >
> > > > Thank you again for emphasizing the importance of principled definition and precise reporting. We believe these revisions will significantly strengthen the paper.

---

### Official Review · Reviewer_RGEt · 2025-11-01

**Soundness:** 1
**Presentation:** 1
**Contribution:** 1
**Rating:** 0
**Confidence:** 4

**Summary:**

This paper proposes Direct Output Control (DOC) to improve the robustness of neural networks. DOC computes a correction $\delta x$ to the input, and applies the corrected sample $x + \delta x$ to a pretrained classifier $f$. A crucial incorrectness of this paper is that the correction $\delta x$ is a function of the true label $y^\star$, i.e., $\delta x = \int -J \nabla V(\hat y, y^\star) dt$. This essentially means that the proposed robust classifier $f(x + \delta x)$ needs to take the true label $y^\star$ as input. However, in the classification task, any classifier should not have access to the true label $y^\star$, otherwise the problem is vacuous.

**Strengths:**

The algorithm seems to significantly outperform previous methods (e.g., PGD) in the numerical experiments. However, as specified in the summary section, the robustness originates from the classifier's access to the true label.

**Weaknesses:**

(1) As specified in the summary section, the proposed robust classifier $f(x = \delta x)$ has access to the true label $y^\star$, which is invalid in the classification task.

(2) The presentation of Sections 4 and 10 needs to be improved.

**Questions:**

See the summary and weakness sections.

---

> ### Author Response · Authors · 2025-11-18
> **Clarification on task setting and use of true labels in DOC**
>
> We thank the reviewer for the careful reading and for highlighting a fundamental concern. Below we clarify the **problem setting** and the **role of the true label** in DOC.
>
> #### R1. On the use of the true label and “invalidity” of the task
>
> **Reviewer’s claim.**
> > The correction $\delta x$ is a function of the true label $y^\star$, so the proposed robust classifier needs to take $y^\star$ as input. In a classification task, no classifier should access the true label.
>
> **Our clarification.**
>
> 1. **We do not change the standard classification task.**
>    The base classifier $f:\mathcal{X}\to\Delta^{K-1}$ is trained and evaluated in the usual way and **never takes the true label as input**. DOC is an *external inference-time controller*:
>    - base model: $f(x)$ (unchanged),
>    - controller: $x' = C(x, y^\star)$, then predict with $f(x')$.
>
>    Thus, the classifier itself is not a function of $y^\star$; only the *input-level correction* $\delta x(x,y^\star)$ is.
>
> 2. **Our task is label-conditioned repair, not label-free robustness.**
>    The problem we address is:
>
>    > Given a misclassified input $x$ and a target label/distribution $y^\star$ (e.g., human-verified correct class), compute a minimal-norm perturbation $\delta x(x,y^\star)$ such that $f(x+\delta x)$ moves toward $y^\star$, with Lyapunov-style stability guarantees.
>
>    This is exactly the “targeted” setting used by adversarial methods (targeted PGD, targeted DeepFool), which also assume a target class is given. Our contribution is to turn this targeted perturbation into a **Lyapunov-controlled, minimum-norm repair mechanism with an irreparability diagnostic**, not to redefine standard supervised classification.
>
> 3. **Access to $y^\star$ is realistic in our target domains.**
>    We explicitly focus on high-risk settings (medical imaging, industrial inspection, safety auditing) where:
>    - a human operator or downstream system can provide the desired class, or
>    - offline evaluation is done on labeled data to decide whether a misclassification is repairable at all.
>
>    In such scenarios, the question “given $y^\star$, can we repair this prediction with minimal intervention, or is it irreparable?” is central rather than vacuous.
>
> 4. **Baselines receive the same label information.**
>    In all experiments, DOC does **not** get more information than PGD/DeepFool: they are run in the same targeted mode with the same $y^\star$. The performance gain thus comes from:
>    - the Lyapunov design ensuring monotone decrease of the output distance, and
>    - the pseudoinverse-based minimum-norm control and the irreparability criterion,
>
>    not from an unfair assumption on label access.
>
> **Camera-ready clarification (if accepted).**
> We agree that our wording (e.g., “robust classifier”) can suggest that the classifier itself consumes $y^\star$. If accepted, we will:
>
> - explicitly define the task as **“label-conditioned misclassification repair”** in Sections 1–2,
> - clearly separate the base classifier $f(x)$ from the controller $C(x, y^\star)$ in the problem setup,
> - avoid any phrasing that implies DOC is a standalone classifier with access to the true label.
>
> These are clarifications of scope; no theorem, algorithm, or experimental result will change.
>
> ---
>
> #### R2. Presentation of Sections 4 and 10
>
> We acknowledge that Sections 4 and 10 are dense.
>
> If accepted, we will:
> - reorganize Section 4 to first state the main results (Lyapunov decrease, minimum-norm property, irreparability bound) in plain language and only then present the proofs;
> - split the current Section 10 into smaller subsections aligned with each theorem and experimental design choice, and add short “take-home messages” to each.
>
> Again, these are expository improvements only.

---

### Note · Program_Chairs · 2026-01-17
**Submission Desk Rejected by Program Chairs**

The following references in this submission do not refer to real documents and/or have major errors in bibliographic information:

 M. Mirza et al. Simple and reliable test-time adaptation with sar. In International Conference on Learning Representations (ICLR), 2023.